# *Leishmania* highjack host lipid body for its proliferation in macrophages by overexpressing host Rab18 and TRAPPC9 by downregulating miR-1914-3p expression

Chandni Sood[1☯], Jitender Kumar Verma[2,1☯], Rituparna Basak[2☯], Anjali Kapoor[2], Swarnima Gupta[2], Amitabha Mukhopadhyay [2,1] *

**1** National Institute of Immunology, New Delhi, India, **2** Kusuma School of Biological Sciences, Indian Institute of Technology, Hauz Khas, New Delhi, India

☯ These authors contributed equally to this work.
* amitabha@bioschool.iitd.ac.in

**Data Availability Statement:** All relevant data are within the manuscript and its Supporting Information files.

## Abstract

Lipids stored in lipid-bodies (LBs) in host cells are potential sources of fatty acids for pathogens. However, the mechanism of recruitment of LBs from the host cells by pathogens to acquire fatty acids is not known. Here, we have found that *Leishmania* specifically upregulates the expression of host Rab18 and its GEF, TRAPPC9 by downregulating the expression of miR-1914-3p by reducing the level of Dicer in macrophages via their metalloprotease gp63. Our results also show that miR-1914-3p negatively regulates the expression of Rab18 and its GEF in cells. Subsequently, *Leishmania* containing parasitophorous vacuoles (Ld-PVs) recruit and retain host Rab18 and TRAPPC9. *Leishmania* infection also induces LB biogenesis in host cells and recruits LBs on Ld-PVs and acquires $FLC_{12}$-labeled fatty acids from LBs. Moreover, overexpression of miR-1914-3p in macrophages significantly inhibits the recruitment of LBs and thereby suppresses the multiplication of parasites in macrophages as parasites are unable to acquire fatty acids. These results demonstrate a novel mechanism how *Leishmania* acquire fatty acids from LBs for their growth in macrophages.

## Author summary

*Leishmania donovani* which causes fatal disease known as visceral leishmaniasis, is dependent on its host to scavenge fatty acids for its survival. But, how *Leishmania* get fatty acids from host macrophages is not known. Lipid bodies are unique organelles that contain neutral lipids and fatty acids, therefore, LBs in macrophages could be the potential source of fatty acids for parasites. We have found that *Leishmania* upregulate the expression of host Rab18 and its GEF by downregulating the expression of miR-1914-3p in infected macrophages via the Ld-gp63-mediated inhibition of dicer expression. As biogenesis of LBs is shown to be regulated by Rab18, we have found that *Leishmania* infection induces LB biogenesis in macrophages and subsequently recruits LBs along with Rab18 and its GEF on *Leishmania* containing parasitophorous vacuoles. Finally, parasites-containing

**Funding:** This work is supported by grants from Department of Biotechnology, Government of India [BT/PR42073/BRB/10/1979/2021] and J.C. Bose Fellowship (SR/S2/JCB-24/2009) to AM by Science and Engineering Research Board, Government of India. CS is supported by fellowship from J.C. Bose Fellowship to AM. JKV (13(9095-A)/2019-Pool), AK (09/086(1373)/2019-EMR-I) and SG (09/086 (0085)/2020-EMR-I) are supported by fellowships from the Council of Scientific and Industrial Research, Government of India. RB (2018BLZ8469) is supported by fellowship from Indian Institute of Technology, New Delhi, India. The funders had no role in study design, data collection and analysis, decision to publish, or preparation of the manuscript.

**Competing interests:** The authors have declared that no competing interests exist.

vacuoles acquire $FLC_{12}$-labeled fatty acids from LBs to meet their nutritional requirement. We have also shown that overexpression of miR-1914-3p in macrophages significantly inhibits the recruitment of LBs and thereby suppresses the growth of parasites in macrophages as parasites are unable to acquire nutrients from LBs. These results demonstrate how *Leishmania* acquire nutrients like fatty acids from LBs for their proliferation in macrophages.

## Introduction

Pathogens use various strategies to acquire essential nutrients from host cells by hijacking host trafficking pathways to ensure their survival. LBs are unique organelles that contain neutral lipids and fatty acids [1,2]. It has been shown that intracellular pathogens like *Mycobacterium*, *Salmonella*, *Coxiella*, *Leishmania* etc. [3–5] containing phagosomes move in close vicinity of LBs in the host cells, but the functional significance and mechanism of their close proximity still remain unknown [6–8].

Incidentally, LB biogenesis has been shown to be regulated by various Rab proteins [9] namely, Rab18, Rab1, Rab5, Rab7, Rab8a and Rab32 in mammalian cells. Among them, Rab18 is the best characterized in regulating the trafficking of LBs from endoplasmic reticulum [10,11]. Similarly, it has been shown that mammalian TRAPPII specific subunits TRAPPC9 and TRAPPC10 act as GEF for Rab18 and activate LB biogenesis [10]. Rab GTPases are master regulators of intracellular trafficking pathways [12–14]. We and others have shown that various intracellular pathogens like *Salmonella* [15–17], *Mycobacterium* [18], *Legionella* [19,20] etc. modulate or mimic the functions of Rab GTPases in host cells for their survival [21,22]. Thus, it is tempting to speculate that *Leishmania* may modulate the function or expression of host Rab18 and its GEF to recruit LBs on PV. In addition, several intracellular pathogens modulate the expression of host miRNAs to modify the functions of host proteins for their advantage [23–26]. Concurrently, we have shown that *Leishmania* avoid their transport to the lysosomes by inducing the expression of host Rab5a by downregulating the expression of miR-494 in infected macrophages [25]. Therefore, it is possible that *Leishmania* might target the host miRNA to modulate the expression of Rab18 and its GEF in macrophages for acquiring essential nutrients from LBs.

Here, we have shown that *L. donovani* upregulate the expression of Rab18 and TRAPPC9 in macrophages by inhibiting the expression of miR-1914-3p which enhances the LB biogenesis in infected cells and consequently recruit them on PV. The enhanced recruitment of host Rab18 and its GEF on Ld-PVs facilitate to acquire LBs and thereby parasites obtain fatty acids from LBs for their proliferation in macrophages.

## Results

### *Leishmania* induce the expression of Rab18 in macrophages and recruit Rab18 on Ld-PVs

As Rab18 predominantly regulates the biogenesis and maturation of LBs in mammalian cells [10,11], we compared the expression of host Rab18 along with other Rabs in *Leishmania* infected and uninfected macrophages by Western blot analysis using specific antibodies. Interestingly, we found that *L. donovani* induce more than 2-fold expression of Rab18 in macrophages after 24 h of infection compared to uninfected control (Fig 1A). Rab5a was used as positive control as we showed previously that *Leishmania* enhance the expression of Rab5a in

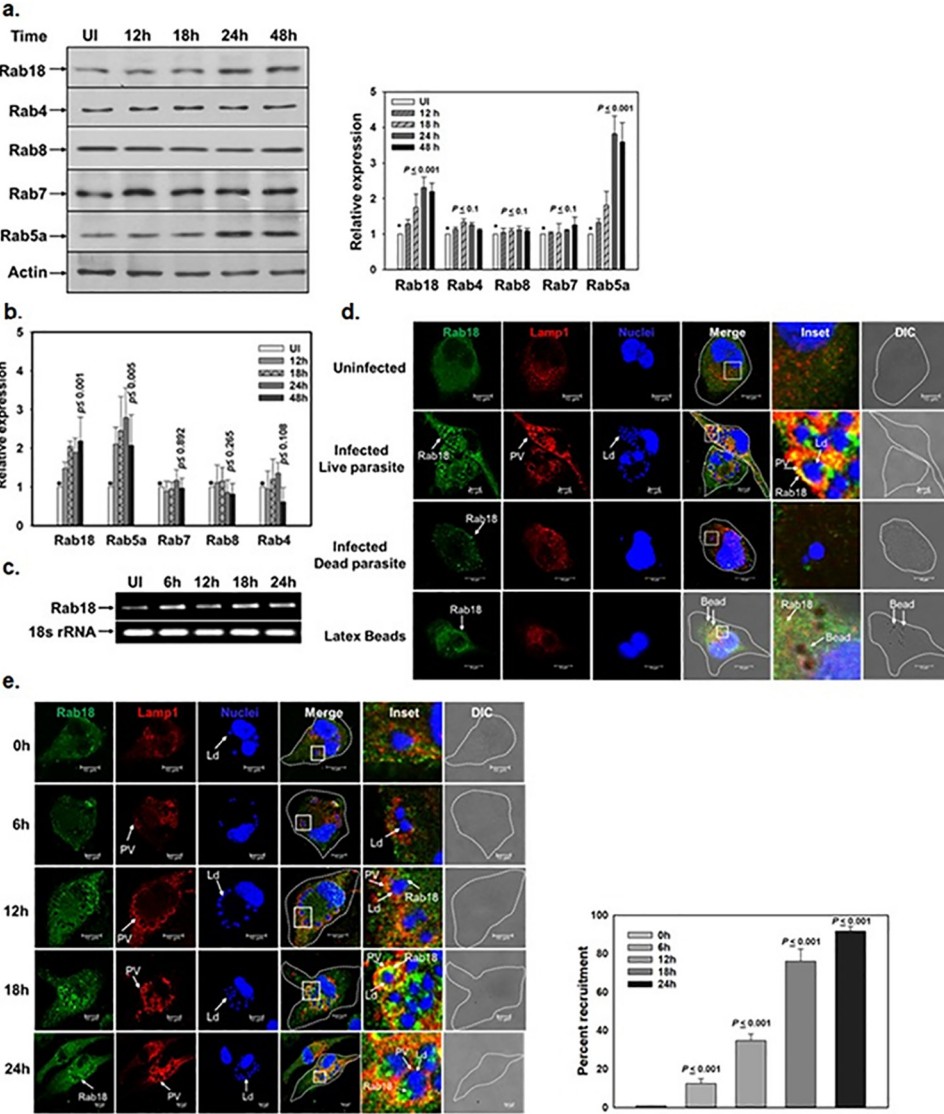

**Fig 1. Expression and recruitment of host Rab18 in *L. donovani* infected macrophages. a,** To determine the levels of different host Rab GTPases in *L. donovani* infected and uninfected macrophages, cells were lysed at indicated time points and Western blot analysis was carried out using specific human antibodies as mentioned in Materials & Methods. Actin was used as loading control. Right panel indicates the quantitation of the respective host Rab proteins. **b,** Levels of different host Rabs in *L. donovani* infected and uninfected differentiated macrophages at respective time points were determined by qPCR as described in Materials & Methods. The respective gene amplification was normalized using 18s rRNA as an internal control. All results are represented as mean ± S.D. of three independent experiments and normalized to uninfected control of respective Rabs arbitrarily chosen as one unit. Results of the indicated groups were analyzed by paired *t* test and levels of significance are indicated by *P* value. **c,** Levels of host mRNA of Rab18 in *L. donovani* infected and uninfected macrophages by limited dilution semi-quantitative RT-PCR. 18s rRNA was used as a control. **d,** Differentiated THP-1 macrophages were infected with *L. donovani* and recruitment of host Rab18 on LAMP1 labelled Ld-PVs were determined by immunostaining after 24 h of infection using specific antibody against human Rab18 (1:50) as described in Materials & Methods. THP-1 cells incubated with dead parasites and latex beads were used as control. **e,** Kinetics of host Rab18 recruitment on Ld-PVs was determined at indicated times as described in Materials & Methods. Right panel indicates the quantitation of host Rab18 recruitment on Ld-PVs at different time points. *Leishmania* and macrophage nuclei were stained with DRAQ5 (Blue). All results are representative of three independent experiments.

infected macrophages [25]. No significant changes in the levels of expression of Rab4, Rab8 and Rab7 in infected macrophages were observed. Subsequently, we determined the transcript levels of different Rabs in host cells by Real-Time PCR (qPCR) and our results also showed about 2-fold induction in the levels of Rab18 and Rab5a transcripts in infected macrophages after 24 h compared to uninfected control cells (Fig 1B). Enhanced expression of Rab18 in *L. donovani* infected macrophages was further confirmed by limited dilution semi-quantitative RT-PCR (Fig 1C). These results indicated that *Leishmania* overproduce Rab18 in macrophages possibly by transcriptional activation.

*L. donovani* not only enhanced the expression of host Rab18 in infected macrophages, but we also found that parasites recruit Rab18 in/on LAMP1 positive Ld-PVs by confocal microscopy (Fig 1D). Recruitment of host Rab18 on PVs required the presence of live parasites as dead parasite containing PV and latex bead containing phagosomes were unable to recruit Rab18 on PVs in macrophages (Fig 1D). However, our results showed that Ld-PV does not recruit Rab7, Rab8 and Rab27 from macrophages (S1 Fig.). Further quantitation revealed that more than 90% of Ld-PVs recruit Rab18 from macrophages whereas less than 10% of PVs recruit other indicated Rabs (S1 Fig). Kinetic analysis of Rab18 recruitment on Ld-PV from macrophages revealed that parasites recruit Rab18 on Ld-PVs in a time dependent manner and retained Rab18 on PVs throughout the experimental period (Fig 1E). About 90% Ld-PVs were found to be positive for Rab18 after 24 h infection in macrophages.

### *Leishmania* overexpress Rab18 specific GEF in macrophages and recruit it on Ld-PVs

Rab GTPases cycle between an inactive GDP-bound form and active GTP bound form. This switch is controlled by guanine nucleotide exchange factors and GTP bound form is recruited on target membrane [14] and promotes transport by fusion between specific donor and acceptor vesicles [27,28]. As Rab18 was found to be recruited on Ld-PVs, we evaluated the levels of Rab18 specific GEF in *L. donovani* infected macrophages. A recent study showed that mammalian TRAPPII complex acts as a GEF for Rab18 [10], therefore, we compared the expression of TRAPPII-specific subunits TRAPPC9 and TRAPPC10 in *L. donovani* infected and uninfected macrophages. Our results showed that *L. donovani* induce the expression of both TRAPPC9 and TRAPPC10 in infected macrophages in a time dependent manner in comparison to uninfected cells (Fig 2A). We also found that parasites induce about 2-fold expression of transcripts of both subunits of TRAPPII complex in infected macrophages compared to uninfected cells (Fig 2B). However, parasite mediated enhanced expression of TRAPPC9 in infected macrophages was more pronounced than TRAPPC10. Subsequently, our results showed that Ld-PVs recruit TRAPPC9 in infected macrophages (Fig 2C). Thus, parasites induce the expression of TRAPPC9 in infected macrophages by transcriptional activation and recruit it on Ld-PVs.

### Mechanism of overexpression of Rab18 and TRAPPC9 by *Leishmania* in infected macrophages

To understand the mechanism of upregulation of Rab18 and TRAPPC9 expression in *L. donovani* infected macrophages, we focused on the modulation of miRNA expression as several intracellular pathogens alter the expression of host miRNAs to establish infection [23–26]. Therefore, we compared the miRNA profile of uninfected and infected macrophages [25]. Further analysis of previously identified miRNA profile data showed that *L. donovani* infection upregulate the expression of 55 miRNAs and 40 miRNAs are downregulated in infected macrophages in comparison to uninfected cells (Fig 3A). Subsequent screening of these miRNAs

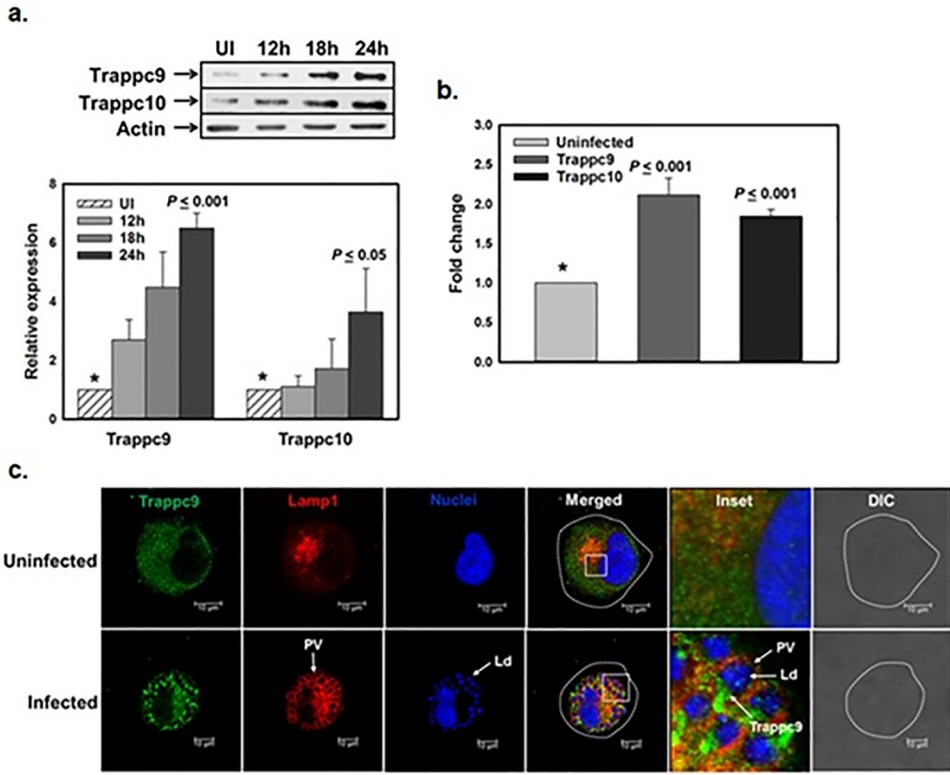

**Fig 2. Expression and recruitment of host TRAPPC9 in *L. donovani* infected macrophages. a,** To determine the levels of host TRAPPC9 and TRAPPC10 in *L. donovani* infected and uninfected macrophages, cells were lysed at indicated time points and Western blot analysis was carried out using specific human antibodies as mentioned in Materials & Methods. Actin was used as loading control. Lower panel indicates the levels of the host Trappc9 and Trappc10 proteins at indicated time points. **b,** Levels of TRAPPC9 and TRAPPC10 in infected and uninfected differentiated macrophages at respective time points were determined by qPCR as described in Materials & Methods. The respective gene amplification was normalized using 18s rRNA as an internal control. All results are represented as mean ± S.D. of three independent experiments and normalized to uninfected control of respective Rabs arbitrarily chosen as one unit. Results of the indicated groups were analyzed by paired *t* test and levels of significance are indicated by *P* value. **c,** Differentiated THP-1 macrophages were infected with *L. donovani* and recruitment of TRAPPC9 on LAMP1 labelled Ld-PVs was determined by immunostaining after 24 h of infection using TRAPPC9 specific antibody. *Leishmania* and macrophage nuclei were stained with DRAQ5 (Blue). All results are representative of three independent experiments.

revealed that the seed region (3′-UGGGGAG-5′) of miR-1914-3p contains 7-mer target site (5′-ACCCCUC-3′) in 3′-UTR of both Rab18 and TRAPPC9 (Fig 3B). Therefore, we determined the level of expression of miR-1914-3p in *L. donovani* infected macrophages. Our results showed about 45% inhibition of miR-1914-3p expression in parasite infected macrophages after 12 h of infection compared to uninfected cells (Fig 3C).

To determine the regulation of Rab18 expression by miR-1914-3p, we made a chimeric construct by ligating 3′-UTR of Rab18 with luciferase as heterologous reporter using pmir-GLO vector. We also made another chimeric construct of 3′-UTR of Rab18 containing mutation in miR-1914-3p recognition element (5′-GAAAAGAC-3′) to determine the specificity. The chimeric construct of 3′-UTR of Rab18 or its mutant was co-transfected with miR-1914-3p mimic into HeLa cells and firefly luciferase reporter activity was measured after 48 h of transfection. Nonspecific miR mimic was used as a control. Our results showed (Fig 3D) that transfection with the miR-1914-3p (40 nM) inhibited about 50% luciferase activity of Rab18 3′-UTR reporter, whereas no inhibition was detected with mutant Rab18 3′-UTR reporter. To understand how miR-1914-3p expression was downregulated in *L. donovani* infected

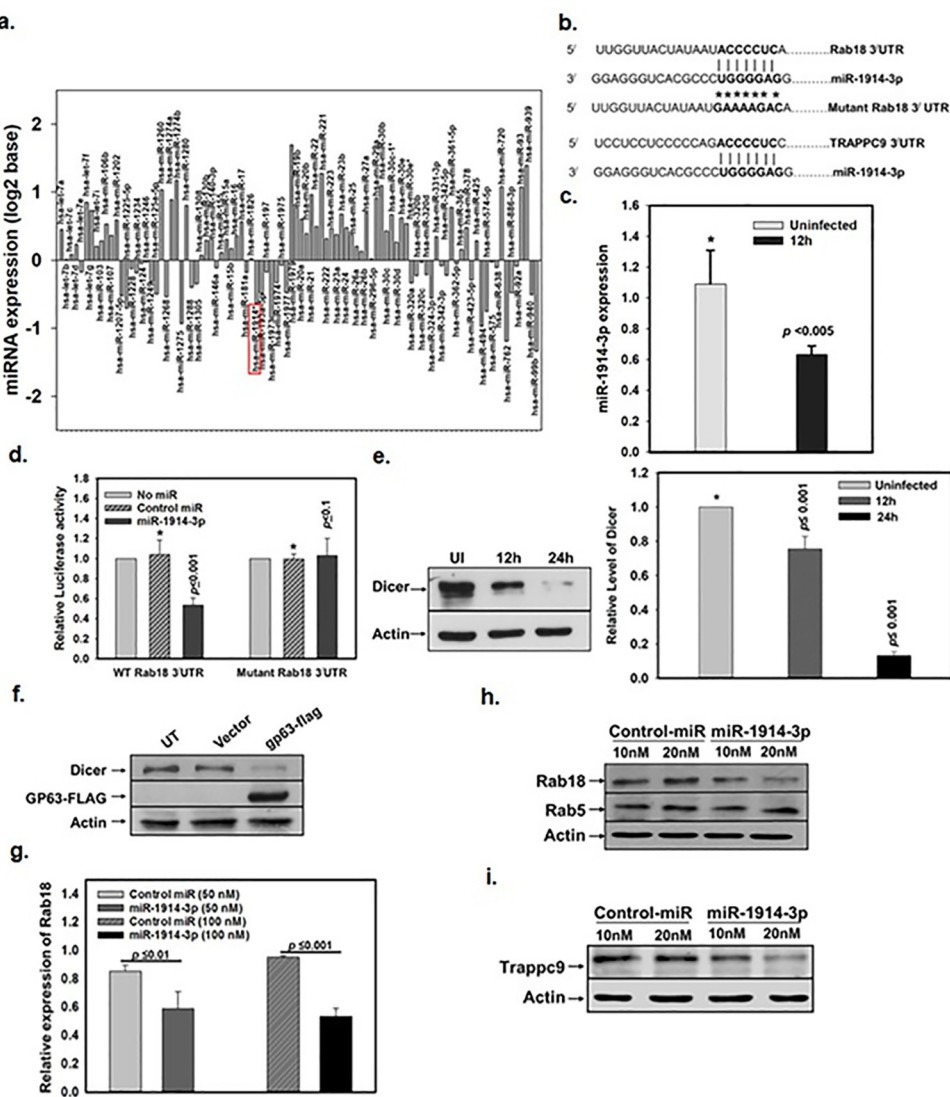

**Fig 3. Mechanism of overexpression of host Rab18 by *Leishmania* in infected macrophages. a,** Modulation of the expression of several host miRNAs in *L. donovani* infected macrophages was revealed by microarray analysis. Relative expression of miRNAs in infected cells was determined in comparison to uninfected control. The whole microarray data have been submitted in Gene Expression Omnibus database (accession number GSE89529). **b,** Sequence alignment of $3'$-UTR of host Rab18 and TRAPPC9 containing 7-mer target site with the seed region of miR-1914-3p predicted by TagetScan tool. Mutation in the target site of $3'$-UTR of Rab18 is highlighted in bold. **c,** Expression of miR-1914-3p in *L. donovani* infected and uninfected THP-1 macrophages was determined 12 h post infection using qPCR as described in Materials & Methods. **d,** miR-1914-3p mediated regulation of Rab18 expression was determined by co-transfecting pmir-GLO chimeric construct containing Rab18 $3'$-UTR or its mutant with miR-1914-3p (40 nM) or control mimic miR into semiconfluent HeLa cells. Luciferase activity was determined from lysed cells after 48 h of transfection as described in Materials & Methods. Results are expressed as relative luciferase activity. Untreated control cells were arbitrarily chosen as one unit. **e,** THP-1 differentiated macrophages were infected with *L. donovani* and expression level of Dicer in infected cells was determined at indicated time points using specific antibody against human Dicer. Uninfected cells were used as control. Actin was used as loading control. Right panel indicates the quantitation of host Dicer level at different time points. **f,** Expression of Dicer was determined in HeLa cells transfected with plasmid containing Flag tagged Ld-gp63 by Western blot analysis using specific antibody against Dicer. Level of gp63 in untransfected and vector transfected cells were also determined by Western blot analysis using anti-Flag antibody. Actin was used as loading control. **g,** Differentiated THP-1 macrophages were transfected with control miR or miR-1914-3p at indicated concentrations and expression of Rab18 was measured by qPCR as described in Materials & Methods. **h,** Levels of host Rab18 and Rab5 proteins in control miR or miR-1914-3p (10 nM or 20 nM) transfected THP-1 cells was determined by Western blot analysis using specific antibodies against human Rab18 and Rab5. Actin was used as loading control. **i,** Similarly, level of TRAPPC9 protein in control miR or miR-1914-3p (10 nM or 20 nM) transfected THP-1 cells was determined by Western blot analysis using human anti-TRAPPC9 antibody. Actin was

used as loading control. All results represented as mean ± S.D. of three independent experiments. Results of the indicated groups were analyzed by paired *t* test and levels of significance are indicated by *P* value.

macrophages, we compared the level of Dicer in infected and uninfected macrophages. We found that *Leishmania* infection reduces the level of Dicer in infected macrophages (Fig 3E). Consequently, our result showed that overexpression of Ld-gp63, a metalloprotease in *Leishmania*, also reduces the Dicer expression in HeLa cells (Fig 3F). To demonstrate the regulation of the expression of Rab18 by miR-1914-3p in macrophages, cells were transfected with indicated concentrations of miR-1914-3p or control miR (S2 Fig) and level of Rab18 was determined by qPCR. We found that miR-1914-3p (100 nM) specifically inhibited about 50% expression of Rab18 mRNA compared to the control (Fig 3G). In addition, we found significant inhibition of Rab18 (Fig 3H) and TRAPPC9 protein (Fig 3I) expression in miR-1914-3p (20 nM) transfected macrophages compared to the control-miR transfected cells.

## Ld-PV specifically recruits LBs in infected macrophages

Rab18 and its GEF play major role in trafficking and biogenesis of LBs from endoplasmic reticulum [10,11], therefore, we analyzed the association of LBs with Ld-PVs in infected macrophages. To determine the recruitment of LBs on Ld-PVs, cells were infected with *L. donovani* promastigotes and LBs were stained with BODIPY 493/503 or LipidTOX 24 h post-infection. Our results showed that LBs stained by BODIPY are scattered throughout the cell cytosol in the uninfected macrophages whereas they are found to be recruited on LAMP1 positive Ld-PVs in infected macrophages (Fig 4A). In contrast, latex bead containing phagosomes were unable to recruit LBs. The recruitment of LBs on Ld-PVs was also confirmed by staining the neutral lipids in LBs in macrophages by LipidTOX (Fig 4B). However, vacuoles containing dead parasites were unable to recruit LBs (Fig 4B). In addition, parasites also acquired Rab18 interacting lipid droplet-associated protein (perilipin) on its PV from infected macrophages (Fig 4C). Further quantitation revealed that about 80% of Ld-PVs recruit LBs whereas only 20% latex bead or dead parasite containing phagosomes are associated with LBs (Fig 4D). We also compared the number of LBs and their size in 50 infected and uninfected macrophages. Our results showed that the average number of LBs in uninfected control cells is 88±41 whereas the average number of LBs in infected cells is found to be 228±111 (Fig 4E). However, average size of LBs in infected ($0.45\pm0.35\mu m^2$) and uninfected ($0.43\pm0.30\ \mu m^2$) cells was similar (Fig 4E). These results indicated that *L. donovani* possibly induce LB biogenesis in the host cells by overexpressing Rab18 and TRAPPC9. Moreover, our results showed that Ld-PVs contain both Rab18 and LBs in infected macrophages (S3 Fig). In addition, we found that calnexin-labelled ER membrane from host cells is also recruited on PVs in infected cells (Fig 4F).

## *Leishmania* containing vacuoles acquire fatty acid from LB for its survival in macrophages

LBs are unique organelles that contain neutral lipids, fatty acids, triglycerides, along with other molecules such as cholesterol, phospholipids [1,2]. To determine the functional significance of recruitment of LBs by Ld-PVs, we analyzed the trafficking of fatty acids (FAs) from LBs to Ld-PVs in the present study because FAs are largely stored in LBs [29]. Therefore, FAs present in LBs in macrophages were pre-labeled with BODIPY-FL-$C_{12}$ ($FLC_{12}$) and then the cells were infected with *L. donovani*. Subsequently, transport of labelled FAs from LBs to LAMP1 positive Ld-PVs was determined. Confocal analysis of *L. donovani* infected macrophages showed that Ld-PVs acquire $FLC_{12}$-labelled FAs (Fig 5A). We also analyzed about 100 Ld-PVs in infected

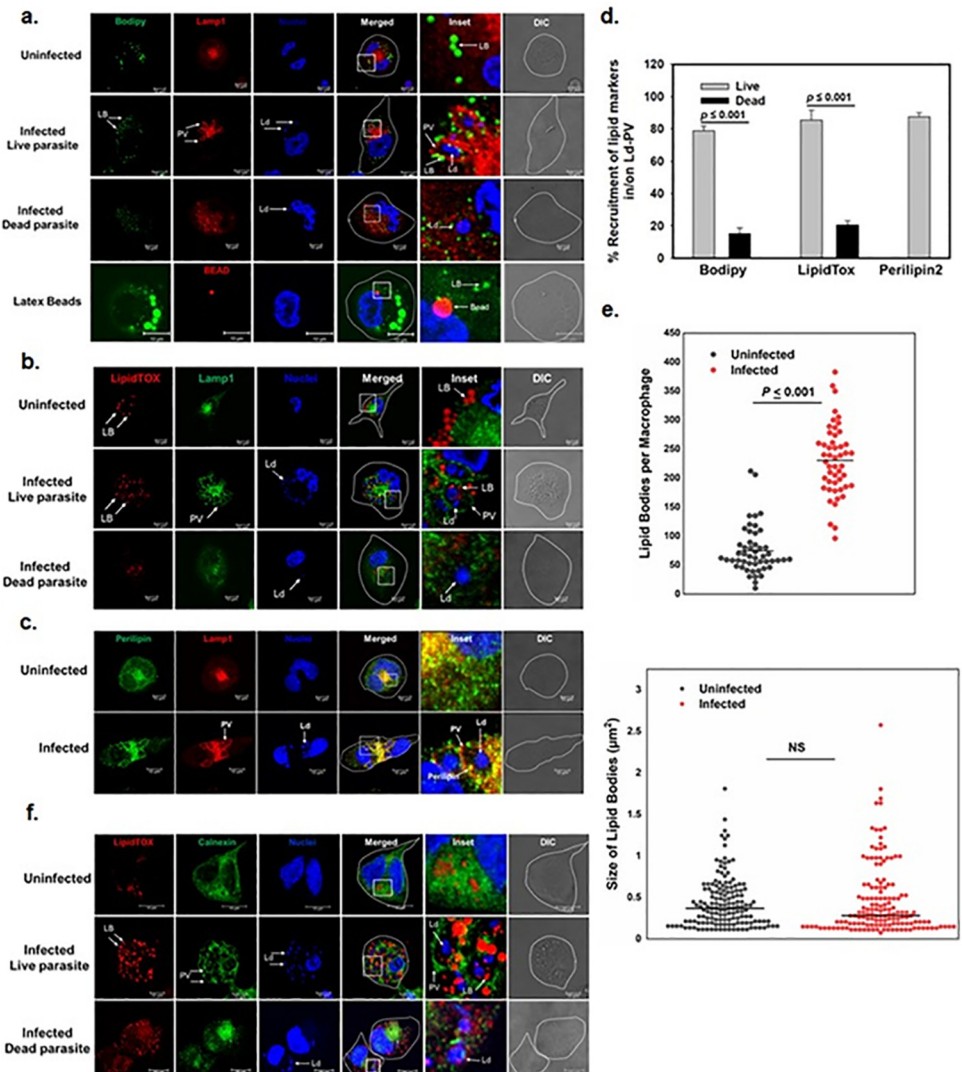

**Fig 4. *Leishmania*-containing PV specifically recruits LBs. a,** Differentiated THP-1 human macrophages were infected with *L. donovani* and recruitment of Bodipy 493/503 stained LBs (Green) by LAMP1 stained Ld-PVs (Red) were detected by confocal microscopy after 24 h of infection as described in Materials & Methods. Uninfected cells and cells incubated with latex beads or dead parasite were used as control. **b,** Recruitment of LipidTOX labelled LBs (Red) by LAMP1 stained Ld-PVs (Green) were determined after 24 h of infection as mentioned in Materials & Methods. Infection with dead parasites was used as control. *Leishmania* and macrophage nuclei were stained with DRAQ5 (Blue). **c,** To determine the recruitment of Perilipin by LAMP1 stained Ld-PVs (Red), Perilipin was immunostained with specific antibody (Green) in *L. donovani* infected and uninfected macrophages as described in Materials & Methods. *Leishmania* and macrophage nuclei were stained with DRAQ5 (Blue). **d,** Percentage of recruitment of different LBs on PVs containing live or fixed parasites were analyzed. All results are represented as mean ± S.D. of three independent experiments. Results of the indicated groups were analyzed by paired *t* test and levels of significance are indicated by *P* value. **e,** Determination of the number and size of LBs in *L. donovani* infected and uninfected macrophages were determined using appropriate software as described in Materials & Methods. Results are representative of mean±SE of 50 macrophages. **f,** Recruitment of ER membrane marker on LB and Ld-PV in *L. donovani* infected macrophages were determined by immunostaining the cells with calnexin specific antibody at indicated time points and analyzed by confocal microscopy. Uninfected cells and cells infected with dead parasite were used as control. *Leishmania* and macrophage nuclei were stained with DRAQ5 (Blue). All results are representative of three independent experiments.

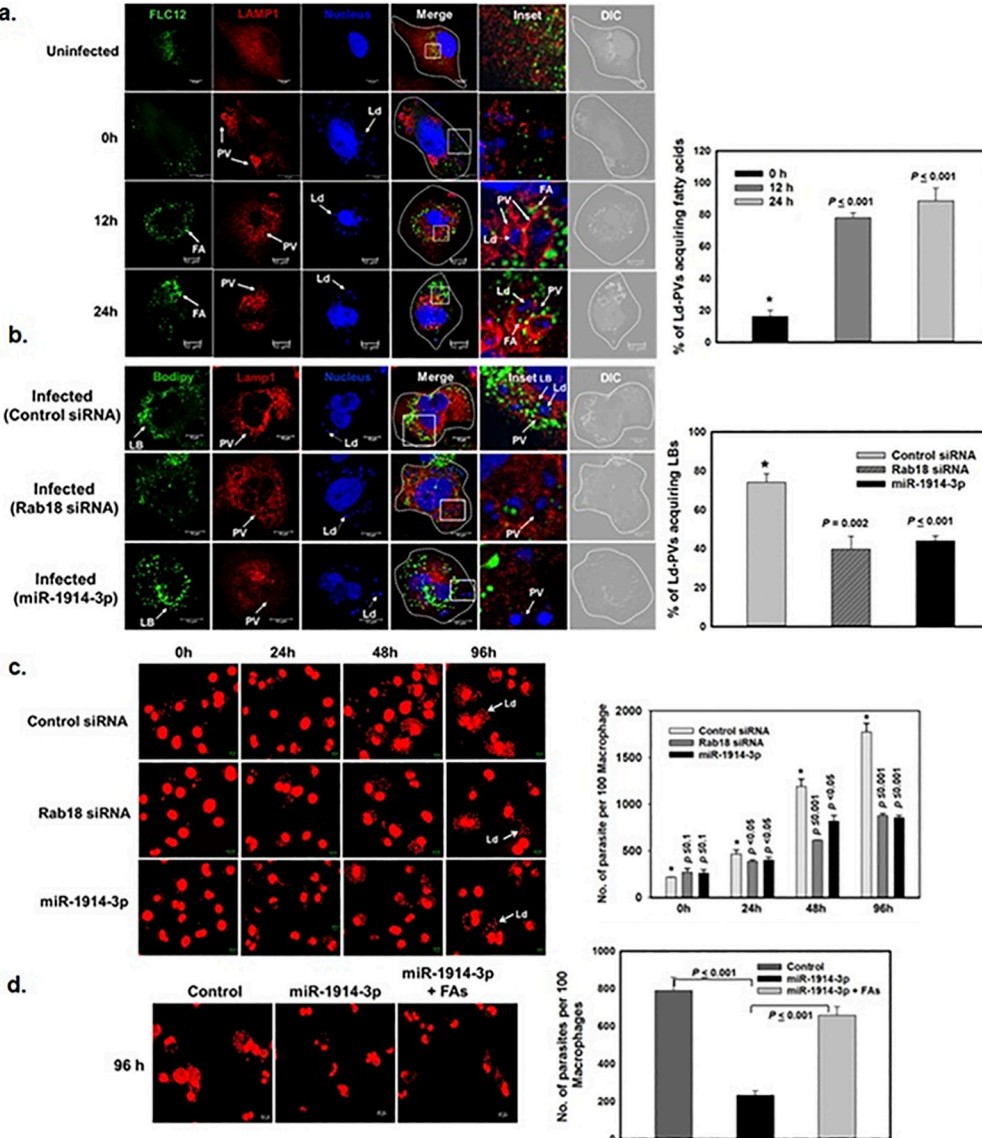

**Fig 5. *Leishmania* acquire fatty acids from LBs for their proliferation in macrophages. a,** Differentiated THP-1 macrophages were incubated with BODIPY-FL-C$_{12}$ to label the fatty acids in LBs and subsequently cells were infected with *L. donovani* as described in Methods. Infected cells prelabeled with BODIPY-FL-C$_{12}$ were chased for indicated time points and appearance of BODIPY-FL-C$_{12}$ (Green) from LBs in Ld-PVs (Red) was determined. Cells were immunostained with LAMP-1 specific antibody to mark the PVs. *Leishmania* and macrophage nuclei were stained with DRAQ5 (Blue). Uninfected cells were used as control. Right panel shows the quantitation of Ld-PVs acquiring BODIPY-FL-C$_{12}$ fatty acids in a time dependent way. **b,** Differentiated THP-1 macrophages were transfected with Rab18 specific siRNA, control siRNA or miR-1914-3p followed by infection with *Leishmania*. Lipid bodies were stained using Bodipy 493/503 as described above. Cells were fixed and immunostained with LAMP-1 to label the Ld-PVs. *Leishmania* and macrophage nuclei were stained with DRAQ5 (Blue). Right panel shows the quantitation of Ld-PVs acquiring Bodipy 493/503 stained LBs under indicated conditions. All results are representative of three independent experiments. **c,** Similarly, miR-1914-3p or Rab18 specific siRNA transfected macrophages were infected with *L. donovani* promastigotes and parasite load in the infected macrophages was microscopically estimated at indicated time. *Leishmania* and macrophage nuclei were stained with propidium iodide (Red). Results are expressed in the right panel as number of parasites present in 100 macrophages ± S.D. from three independent experiments. Results of control (*) and Rab18 siRNA/miR-1914-3p overexpressed cells were analyzed by paired *t* test and levels of significance are indicated by *P* value. **d,** Similarly, miR-1914-3p transfected and *L. donovani* infected macrophages were incubated in the presence or absence of fatty acids supplements. Parasite load in the infected macrophages was estimated at 96 h post infection. Untransfected infected cells were used as control. Results are expressed in the right panel as number of parasites present in 100 macrophages ± S.D. from three independent experiments. Results were analyzed by paired *t* test and levels of significance are indicated by *P* value.

macrophages and determined the presence of $FLC_{12}$-labelled FAs in/on the LAMP1 stained Ld-PV at different time points. Our results showed that about 90% of Ld-PVs acquired $FLC_{12}$-labelled FAs after 24 h in infected macrophages (Fig 5A). To evaluate whether Ld-PV acquire FAs from LBs by overexpression of Rab18 via down-regulation of miR-1914-3p, we determined recruitment of LBs in miR-1914-3p overexpressed macrophages as well as in Rab18 knock-down cells by specific siRNA (S4 Fig). Our results showed that transfection of Rab18 specific siRNA or overexpression of miR-1914-3p specifically downregulated the expression of Rab18 in macrophages (Fig 3H) and significantly inhibited the recruitment of LBs labelled with BODIPY 493/503 (Fig 5B) or perilipin (S5 Fig) on Ld-PVs in macrophages. To determine the function of Rab18 in the proliferation of *L. donovani* in macrophages, Rab18 knock-down or miR-1914-3p overexpressed cells were infected with *Leishmania* and parasite load was determined at indicated time points. Our results showed that parasite infection in macrophages were not compromised in these cells as similar number of parasites were observed in both Rab18 knock-down and miR-1914-3p overexpressed cells at 0 h. However, more than 50% inhibition of parasite load was observed in miR-1914-3p (50 nM) overexpressed macrophages in comparison to control cells after 96 h of infection (Fig 5C). Similar results were also obtained in Rab18 knock-down macrophages by siRNA under identical conditions (Fig 5C). Interestingly, we found that miR-1914-3p mediated inhibition of parasite growth in macrophages is significantly reversed when infected macrophages are incubated in medium containing fatty acids supplements like oleic and linoleic acids (90 ng/ml each) for 96 h (Fig 5D).

## Discussion

*Leishmania donovani* is a protozoan parasite that causes fatal disease known as visceral leishmaniasis that affects annually about 12 million people worldwide [30]. Drugs used for chemotherapy of leishmaniasis are toxic and no licensed vaccine is available [31]. Thus, the major thrust is to identify new chemotherapeutic targets against *Leishmania*. *Leishmania* do not have fully characterized biosynthetic pathways for heme [32,33], purine [34], folate [35], fatty acids [36–38] and depend on the host to scavenge these nutrients for their intracellular survival [39–42]. Thus, understanding the mechanism of acquisition of essential nutrient like fatty acids, heme, folate etc. by the parasite is an attractive area to identify the chemotherapeutic target [43].

Intracellular pathogens have evolved various strategies to acquire essential nutrients from host cells for their survival by manipulating host trafficking pathways [21,22]. It is emerging from recent studies that intracellular pathogens also alter host lipid and carbohydrate metabolic pathways [44,45]. LBs are highly dynamic cytoplasmic organelles which are composed of hydrophobic core of neutral lipids surrounded by a phospholipid monolayer [1,2]. Incidentally, large numbers of intracellular pathogens like *Mycobacterium*, *Salmonella*, *Toxoplasma*, *Coxiella*, *Leishmania*, *Trypanosoma* etc. are found to be in close association with host LBs [3,46,47]. However, mechanism of their association with host LBs by phagosomes containing intracellular pathogens and its functional significance are not well established. Here, we have tried to understand how Ld-PVs recruit LBs to acquire essential nutrients like fatty acids from the host cells for the proliferation of parasites in macrophages.

As biogenesis and maturation of LBs in mammalian cells is predominantly regulated by Rab18 and its GEF [10,11], thus, we have compared the expression of Rab18 along with other Rabs in *L. donovani* infected and uninfected macrophages. Interestingly, we have found that the expression of host Rab18 is specifically upregulated in *L. donovani* infected macrophages by transcriptional activation as higher level of Rab18 transcript is detected by qPCR in infected macrophages. Similarly, we have found that *L. donovani* induce the expression of Rab18

specific GEF, TRAPPII subunits TRAPPC9 and TRAPPC10 in infected macrophages. However, enhanced expression of TRAPPC9 by *L. donovani* in infected macrophages is found to be more pronounced than TRAPPC10, therefore, we have further analyzed the mechanism of upregulation and recruitment of TRAPPC9 in infected cells. *L. donovani* not only overproduce Rab18 and TRAPPC9 in infected macrophages, but parasites also specifically recruit them on LAMP1 positive Ld-PVs and retain them throughout experimental period of 24 h. Taken together, these results suggest that Ld-PVs possibly recruit GTP form of Rab18 from infected cells as Rab GTPases are predominantly associated with target membrane in GTP form [12].

To understand how *L. donovani* induce the expression of Rab18 and TRAPPC9 in infected macrophages, we have analyzed miRNA profiles in *L. donovani* infected and uninfected macrophages as we have reported previously [25]. Interestingly, we have found that miR-1914-3p has target site in $3^{/}$-regulatory region of Rab18 and TRAPPC9 in mammalian cells. Subsequently, we have shown that transfection of miR-1914-3p specifically inhibits the expression of Rab18 and TRAPPC9 in cells. These results are confirmed by using chimeric construct containing $3^{/}$-UTR of Rab18 with luciferase as heterologous reporter. These results indicate that miR-1914-3p binds with recognition element present in the $3^{/}$-UTR of Rab18 and TRAPPC9; thereby, negatively regulates the expression of Rab18 and TRAPPC9 in macrophages. Subsequently, we have addressed how *L. donovani* upregulates the expression of Rab18 and TRAPPC9 in infected macrophages. Our results have shown that *L. donovani* infection downregulates the expression of miR-1914-3p by reducing the expression of Dicer in infected macrophages via its Ld-gp63, a metalloprotease in *Leishmania* [48]. These results are supported by previous studies that *L. donovani* targets Dicer to downregulate the expression of miR-122 [49]. Moreover, it has been shown previously that Ld-gp63 secreted by *Leishmania* into host cells hijacks macrophage signaling by degrading several critical serine/threonine kinases and several transcription factors to alter gene expression [48]. Thus, *L. donovani* infection reduces the expression of Dicer in infected macrophages possibly by degrading it via their Ld-gp63. In addition, it has been shown that silencing of Dicer reduces the level of miR-1914-3p expression in cancer cells [50]. As we and others have shown that *L. donovani* infection reduces the expression of Dicer in the host cells; thus, *Leishmania* infection may downregulate the expression of other Dicer-dependent miRNAs in infected cells. This is supported by the fact that *L. donovani* infection downregulate the expression of 40 other miRNAs including miR-494 which parasite uses to upregulate the expression of Rab5a in infected macrophages [25]. We have also found that parasite upregulates the expression of about 55 miRNAs in infected macrophages. The expression of these miRNAs may be regulated by non-canonical miRNA biogenesis pathways which are possibly independent of Drosha/DGCR8 and Dicer pathways [51,52]. Currently, we are evaluating the role of other miRNAs which are downregulated in *L. donovani* infected macrophages. Consistent with our finding, it has been shown that *Leishmania* infection downregulates miR-122 expression to lower serum cholesterol [49] and overexpress Rab5a in infected macrophages by downregulating the expression of miR-494 [25]. Taken together, our results indicate that *L. donovani* infection reduces the expression of Dicer by Ld-gp63 which suppresses the expression of miR-1914-3p to upregulate the levels of host Rab18 and TRAPPC9 in infected macrophages. Our results also suggest that upregulation of Rab18 and TRAPPC9 induce the biogenesis of LBs in infected cells. These results are supported by previous studies that LB biogenesis in cells is regulated by Rab18 and its GEF [10,11].

The fatty acid biosynthetic pathway in *Leishmania* is not fully characterized [36–38]. Since *Leishmania* belongs to the trypanosomatid family, it is predicted that this parasite may also possess the ability of *de novo* biosynthesis of fatty acids. However, some molecules of this pathway present in *Trypanosoma* are absent in *Leishmania* [37]. Though, *Leishmania* encode for

type II fatty acid synthase (FASII) as well as the fatty acid elongase (FAE) but their role in fatty acid biosynthesis in *Leishmania* is not yet characterized [53]. Thus, *Leishmania* depends on its host to scavenge fatty acids for its intracellular survival [42,54]. As LBs are major source of lipids and FAs [1], therefore, we have analyzed the recruitment of LBs on Ld-PVs. We have found that BODIPY 493/503 and LipidTOX stained LBs from macrophages are recruited on Ld-PVs and recruitment of LBs by PVs require the presence of live parasites. Previous studies have shown that Rab18 regulates the association of LBs with ER membranes [55] and also mediates the direct interactions between LBs with mitochondria and peroxisome [56]. In addition, Rab18 also regulates the translocation of LBs from ER by binding with lipid droplet-associated protein, perilipin-2 [57]. We have found that Ld-PVs also acquire perilipin which may facilitate the interaction of LBs with Ld-PVs by Rab18. Recent studies have shown that the GTP form of Rab18 interacts with the NAG-RINT1-ZW10 tethering complex and their associated SNAREs (Syntaxin18) to mediate the fusion between ER and LBs [11]. Moreover, mammalian TRAPPII complex, consisting of TRAPPC9 and TRAPPC10 subunits, activates Rab18 by acting as GEF and recruits it on the LB surface [10]. We have found that parasites also induce expression of TRAPPC9 and TRAPPC10 subunits of TRAPPII complex in infected macrophages. Thus, it is possible that *Leishmania* infection catalyzes the activation of Rab18 to GTP form in infected macrophages and recruits Rab18 on Ld-PVs. Our results have shown that Ld-PVs also recruit calnexin-labelled ER membrane. Previous studies have shown that Ld-PVs acquire ER membrane by fusion through Sec22b regulated process [58]. Hence, the recruitment of LBs by Ld-PVs may be mediated by similar process as LBs bear the calnexin-labelled ER membrane. Moreover, it has been shown that Rab18 serves as a contact point and facilitates the fusion of ER membrane with LBs [59]. Thus, our results suggest that LBs are possibly recruited on Ld-PVs while obtaining new membrane from ER during maturation of PVs.

Fatty acids are not freely available in cell cytoplasm as they are mainly stored as triglycerides in LBs. Therefore, we have investigated whether Ld-PVs acquire fatty acids from LBs using well established method to determine the transport of fatty acids from LBs to the target organelles by labelling the FAs in LBs with $FLC_{12}$ [29,60]. Therefore, we have first labelled the FAs in LBs in macrophages by $FLC_{12}$ followed by infection of the cells with *Leishmania*, and analyzed the recruitment of $FLC_{12}$-labelled FAs by Ld-PVs. Interestingly, we have found that $FLC_{12}$-labelled FAs are recruited in/on LAMP1 stained Ld-PVs. Similarly, it has been shown that *Toxoplasma* acquires $FLC_{12}$-labelled FAs in their PV from LB by Rab7 mediated activation of lipophagy [29]. However, our results have shown that *Leishmania* enhance the expression of host Rab18 and its GEF in infected macrophages to induce LB biogenesis in the host cells and subsequently Ld-PVs recruit LBs possibly to acquire $FLC_{12}$-labelled FAs by the parasites. Previous studies have shown that single amastigote of *L. donovani* resides in individual PV tightly wrapped by the PV membrane [58,61–63]. Thus, it is possible that $FLC_{12}$-labelled FAs acquired by LdPVs are sequestered in the parasites. Moreover, our results are also supported by the fact that trafficking of FAs from LBs to mitochondria and peroxisomes also requires Rab18 [56,60]. In addition, several other intracellular pathogens accumulate LBs near their phagosomes/PV but their functional significance is not clearly demonstrated [3–5]. Nevertheless, lipidome analysis of intracellular *Trypanosoma cruzi* and their mammalian host cells reveals that the FAs signatures in the triacylglycerol pools are very similar between them indicating parasite may acquire fatty acids from host triacylglycerol [64] by some unknown mechanism. Interestingly, *Mycobacterium* induces the expression of miR-33 in infected macrophages to inhibit autophagy and modulates host lipid metabolism for its intracellular survival [65]. Furthermore, *Mycobacterium* [4] and *Salmonella* [5] containing phagosomes are also found in close association with LBs in host cells possibly to suppress host immune response. Thus, it is

possible that *Leishmania* may also deactivate host immune response by recruiting LBs. As LBs also contain other molecules, such as cholesterol, phospholipids, and even some proteins, it may be possible that Ld-PVs also salvage other nutrients from LBs which need to be evaluated in future studies. We have also shown that selective depletion of Rab18 by specific siRNA or overexpression of miR-1914-3p in macrophages significantly inhibits the recruitment of LBs in infected cells and considerably suppresses the proliferation of intracellular amastigotes in macrophages. Interestingly, we have found that addition of fatty acids supplements like oleic and linoleic acids in the growth medium significantly reversed miR-1914-3p mediated inhibition of parasite growth in macrophages suggesting that parasites possibly acquire fatty acids from LBs. These results indicate that enhanced expression of Rab18 via the inhibition of miR-1914-3p by *L. donovani* is essential to scavenge the FAs from host LBs for the optimal growth of the parasites in macrophages.

In conclusion, this is the first demonstration that *L. donovani* enhances the expression of host Rab18 and its GEF in infected macrophages to recruit LBs for scavenging essential nutrients like FAs from LBs for its intracellular growth. To understand the mechanism of LB recruitment, we have shown that *Leishmania* upregulate the expression of Rab18 and TRAPPC9 in infected macrophages by inhibiting the synthesis of miR-1914-3p via Ld-gp63 mediated reduction of Dicer expression. Enhanced expression of Rab18 and its GEF facilitates LB biogenesis in infected macrophages and subsequently parasites recruit LBs on Ld-PVs to acquire FAs for its proliferation in host cells (Fig 6). These results also suggest the possibility of modulating host cell Rab18 or TRAPPC9 expression by miR-1914-3p or other small molecules to block the acquisition of FAs from LBs which may be exploited for developing potential therapeutic application against leishmaniasis.

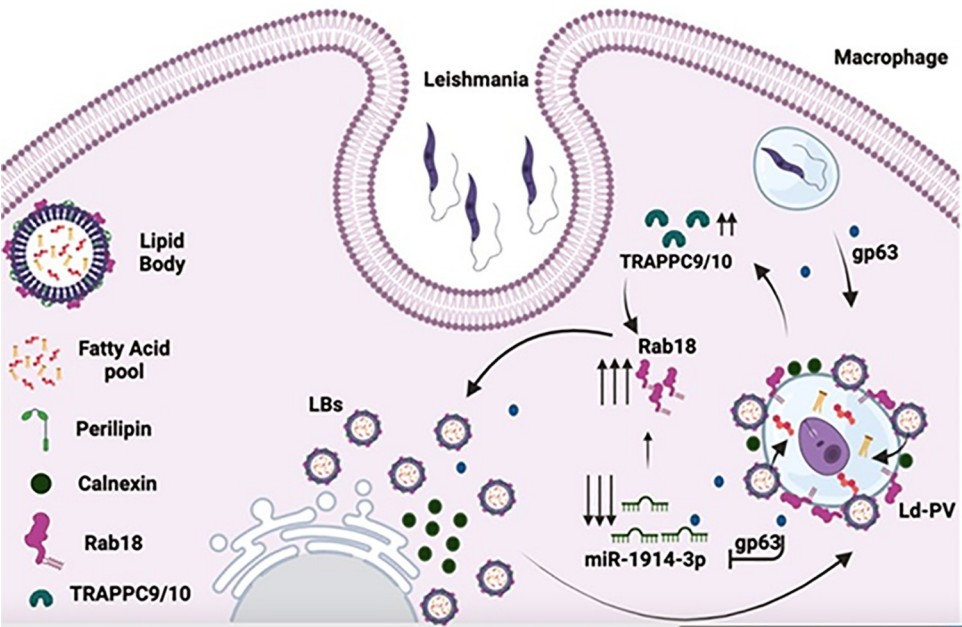

**Fig 6. Schematic representation of mechanism of acquiring fatty acids from LBs by *Leishmania* in macrophages.** *L. donovani* after entering into human macrophages secretes gp63 which degrades dicer to downregulate the expression of host miR-1914-3p and thereby induces the expression of host Rab18 and TRAPPC9 in the infected cells. Thus, parasites trigger LB biogenesis in the infected cells. Subsequently, Ld-PVs recruit LBs along with ER membrane and acquire fatty acids from LBs for their proliferation in macrophages.

## Materials and methods

### Materials

Unless otherwise stated, all reagents were obtained from Sigma Chemical Co. (St.Louis, MO). Tissue culture supplies were obtained from the Grand Island Biological Co. (Grand Island, NY). Lipofectamine 3000 and Lipofectamine RNAi max reagent were purchased from Thermo Fisher Scientific. pmirGLO Dual-Luciferase miRNA Target Expression Vector and Dual-Luciferase Reporter Assay System were purchased from Promega Life Science (Madison, WI). Anti-Rab18 antibody and microRNA mimics were obtained from Sigma Aldrich (St. Louis, MO). Antibodies against Perilipin, Rab5a, Rab27 were purchased from Abcam (Cambridge, England). Anti-actin antibody was obtained from Santa Cruz Biotechnology (Santa Cruz, CA). Antibodies against Lamp1, Rab7, Rab9 and Dicer were obtained from Cell Signalling Technologies (Danver, MA). Anti-Rab4 and anti-Rab8 antibodies were acquired from BD Biosciences, USA. All HRP-conjugated secondary antibodies were purchased from Jackson Immuno-Research Laboratory (West Grove, PA) and ECL was obtained from Amersham Biosciences, UK. Alexa labelled secondary antibodies for immunofluorescence studies, BODIPY 493/503 and LipidTOX were purchased from Molecular Probes (Eugene, OR). Linolenic acid-oleic acid-albumin (100x) was obtained from Sigma Aldrich (St. Louis, MO). All other reagents used were of analytical grade.

### Cells

Human acute monocytic leukemia-derived cell line (THP-1) was obtained from American Type Culture Collection, Manassas, Virginia. Cells were routinely cultured in complete RPMI (RPMI-1640 containing 10% FCS and 50 μg/ml gentamycin) at 37˚C in a humidified incubator with 5% $CO_2$. THP-1 cells were differentiated into macrophages in the presence of Phorbol 12-myristate 13-acetate (100 ng/ml) for 24 h. Cells were washed and incubated for another 24 h in complete RPMI without PMA and used for all experimental procedures.

### Infection

Differentiated THP-1 human macrophages were infected with *L. donovani* promastigotes as described previously [25]. *Leishmania donovani* Bob strain (LdBob strain/MHOM/SD/62/1SCL2D) was used for all experiments. To maintain the virulence, parasites were repeatedly passaged in BALB/c mice. Parasite culture was maintained in complete M199 medium supplemented with HEPES (40mM) containing FCS (10%), Penicillin-Streptomycin (100U/ml) and Hemin (10 μg/ml) at 23˚C. THP-1 differentiated human macrophages were infected with *Leishmania* promastigotes in serum free RPMI medium at multiplicity of infection (20:1; parasite/macrophage) for 3 h at 37˚C followed by 3 washes with plain RPMI to remove uninternalized parasites. Infected cells were incubated in complete RPMI at 37˚C for indicated time points.

### Immunofluorescence microscopy

For immunofluorescence studies, *L.donovani* infected or uninfected cells were fixed with 4% PFA (for 20 min) or ice-cold methanol (10 min). After fixation, cells were washed with PBS and permeabilized with permeabilization buffer (PBS containing 0.1% Triton X-100 for 10 min) at 24˚C. Subsequently, cells were blocked with blocking buffer (PBS containing 3% BSA, 0.1% Triton X-100, 10 μg/ml RNase A) for 2 h at 37˚C. Permeabilized cells were further probed with specific antibodies against indicated protein in blocking buffer for 1 h at 24˚C. Finally, cells were washed three times with PBS and incubated with Alexa Fluor labelled secondary

antibody (1:1000) for 1 h at 24°C in blocking buffer. Propidium iodide (1:50) or Draq5 (1:100) was used to label the host and parasite nuclei. Cells were mounted in Prolong gold antifade mounting reagent and viewed in Leica confocal microscope (DMi8) using an oil immersion 63X objective. A sequential scan was done for visualizing two or more fluorophores within the same sample and similar settings were used for different replicates of the same experiment.

## Lipid bodies staining

Lipid bodies in infected and uninfected cells were stained using Bodipy 493/503 or LipidTOX as per manufacturer's protocol. Briefly, *Leishmania* infected and uninfected THP-1 cells were fixed using 4% paraformaldehyde (PFA) at 24°C for 20 min. Cells were permeabilized using 0.1% Saponin and incubated with Bodipy 493/503 (15 nM for 30 min) or LipidTOX (1:500 for 20 min) at 24°C. To check the trafficking of fatty acids in *Leishmania*, THP-1 cells were prelabelled with BODIPY $FLC_{12}$ and subsequently infected with *Leishmania*. Briefly, differentiated THP-1 cells were incubated with BODIPY $FLC_{12}$ (1 μM) for 18 h and subsequently infected with *Leishmania* promastigotes (20:1; parasite/macrophage) for 3 h at 37°C as described in previous section. Infected cells were incubated in complete RPMI at 37°C for indicated time points. After washing the cells, parasites nuclei were stained by Draq5 (1:100) and PVs were immune-stained using anti-LAMP1 (1:50) antibody. Coverslips were mounted in ProLong gold antifade reagent (Molecular Probes) and viewed in a Leica confocal microscope (Leica DMi8) using an oil immersion objective (63X).

## Detection of levels of various proteins in infected and uninfected cells

To detect the levels of various proteins in *Leishmania* infected and uninfected cells, cells were lysed using cell lysis buffer (20 mM Tris-HCl, pH 7.5, 150 mM NaCl, 1 mM $Na_2EDTA$, 1 mM EGTA, 1% NP-40 and 1% sodium deoxycholate) containing protease inhibitors cocktail for 60 min at 4°C and centrifuged at 15000 x g for 15 min. Subsequently, cellular proteins (40 μg) were resolved on a 12% SDS–PAGE and Western blot analyses were carried out using specific antibodies and probed with respective HRP-labeled secondary antibodies. All blots were quantified using ImageJ software.

## Detection of the expression of various proteins in infected and uninfected macrophages by Real-time PCR

To compare the levels of different Rab GTPases and Rab18 specific GEF in infected and uninfected human macrophages, cells were lysed using TRIzol reagent and total RNA was isolated using standard procedure. Subsequently, cDNA was synthesized using Thermo Script RT-PCR kit (Invitrogen) according to the manufacturer's instructions. Real-time PCR (qPCR) was carried out using respective specific forward and reverse primers and 18s rRNA was used as an internal control. The final volume of reaction mixture for qPCR was 35 μl (17.5 μl of SYBR Green, 3 μl of cDNA and 1 μl of each forward and reverse primer). The samples were run in triplicate in an ABI 7500 Fast Real-Time PCR detection system using following thermal conditions: 50°C for 20 sec followed by 10 min at 95°C, then 40 cycles at 95°C for 15 sec and 60°C for 1 min. The results were analyzed using comparative Ct method ($2^{-\Delta\Delta Ct}$). The respective gene amplification was normalized using 18s rRNA as an internal control and expressed as relative unit compared to uninfected control arbitrarily chosen as 1 unit. Primer sequences used in qPCR are indicated below

a.  hRab18: Forward 5$^/$- GAGGCAAGTGCAAAAACCTG-3$^/$ and Reverse 5$^/$- CACAGAGCA ATAACCACCACAG -3$^/$;

b. hRab5a: Forward 5$^{/}$-TATTGGCCCCTTGAATTCTG-3$^{/}$ and Reverse 5$^{/}$-TTAGAAAAGCA GCCCCAATG-3$^{/}$;

c. hRab7: Forward 5$^{/}$-GCGGAGCTTTTTCCTCTTTT-3$^{/}$ and Reverse 5$^{/}$-TTTTATTGGCATT GCGTTCA-3$^{/}$;

d. hRab4: Forward 5$^{/}$-AGGACCTGGATGCAGATCGT-3$^{/}$ and Reverse 5$^{/}$-TCCCCTGTGAG CGCACTT-3$^{/}$;

e. hRab8: Forward 5$^{/}$-TCCTGTTCCGCTTCTC-3$^{/}$ and Reverse 5$^{/}$ GCTGTGTCCCATATCT G -3$^{/}$;

f. hTRAPPC9: Forward 5$^{/}$-TCGCTGGATCCCAAAGATTT-3$^{/}$ and Reverse 5$^{/}$- TCTAGGCT TTGGGCCACATC -3$^{/}$;

g. hTRAPPC10: Forward 5$^{/}$-AGCTGTCTGGAGGTGTTGCA-3$^{/}$ and Reverse 5$^{/}$- CCCAAAC CTGCCAAAAGGT -3$^{/}$;

h. h18s rRNA: Forward 5$^{/}$-CGAAAGCATTTGCCAAGAAT-3$^{/}$ and Reverse 5$^{/}$-AGTCGGCA TCGTTTATGGTC-3$^{/}$

## Microarray analysis of infected and uninfected macrophages

Microarray analysis was previously done using total RNA isolated from *Leishmania* infected and uninfected THP1 differentiated macrophages from commercial facility (Genotypic Technology, Bengaluru, India) as described [25]. Differential expression was further analyzed to determine relative expression of miRNAs in infected cells in comparison to uninfected control.

## Determination of miR-1914-3p level in infected and uninfected macrophages

To determine the expression levels of miR-1914-3p *in L. donovani* infected and uninfected cells, qPCR was done using TaqMan miRNA assays (Life Technologies, 4427975). Briefly, miRNAs were reverse transcribed to cDNA using TaqMan miRNA Reverse Transcription kit (Life Technologies, 4366596). The qPCR was carried out using master mix containing 10 μl of TaqMan Universal PCR master mix, 1 μl of the miRNA specific primers, 1.33 μl of cDNA, and final volume was adjusted to 20 μl with nuclease-free water. U6 snoRNA amplification was used as the internal control (U6 small nucleolar RNA, Life Technologies, 4427975). The PCR was carried out in triplicates in an ABI 7500 Fast Real-Time PCR detection system using following thermal conditions: 10 min at 95˚C followed by 40 cycles at 95˚C for 15 sec and extension for 1 min at 60˚C. Results were analyzed using comparative Ct method ($2^{-\Delta\Delta Ct}$) and expressed as relative expression compared to uninfected control arbitrarily chosen as 1 unit.

## Determination of the regulation of expression of Rab18

To determine the regulation of expression of host Rab18, we cloned the 3$^{/}$ regulatory region of human Rab18 in pMIR-GLO vector and measured the expression of luciferase after transfecting the chimeric construct in cells as described previously [25]. Briefly, the 3$^{/}$ untranslated regulatory region (UTR) (500 bp) of human Rab18 was PCR amplified from cDNA synthesized from THP-1 macrophages by using appropriate forward (5$^{/}$- GTGAGCTCATGTGTGAGG-GAAAATACTG-3$^{/}$) and reverse primer (5$^{/}$- GTCTC GAGAGTAGTAAATCTTGTATATT-TAGTAAAAGC-3$^{/}$) and cloned in pMIR-GLO vector in *SacI/XhoI* restriction sites. In order

to validate the role of miR-1914-3p binding to its predicted binding site in the 3$^{/}$-UTR of Rab18, a mutant UTR lacking miRNA binding site was cloned by megaprimer based approach. To generate the mutant UTR containing desired mutations in the full-length background, a megaprimer was made using WT forward and mutant reverse primer (5$^{/}$-AATACTGGAATCTTACG TCTTTTCATTATAGTAAC-3$^{/}$). A second round of PCR amplification was carried out to amplify the full-length amplicon with desired mutation using mutant megaprimer as forward primer and UTR specific WT reverse primer. Mutant UTR was also cloned in pMIR-GLO vector as described above. To determine miR-1914-3p mediated regulation of hereterologous expression of Rab18 chimeric construct, these chimeric reporter constructs (300ng) were co-transfected with indicated concentration of miR-1914-3p or control mimic miR into semiconfluent HeLa cells using Lipofectamine 3000 reagent (Invitrogen) as per manufacturer's protocol. After 48 h, cells were lysed in 1X passive lysis buffer (Promega) and the lysates were used to measure Firefly and Renilla luciferase activity by Dual-Luciferase reporter assay kit (Promega). Renilla luciferase activity was used as a control reporter for normalization. Results were expressed as relative luciferase activity compared to untreated control cell arbitrarily chosen as 1 unit.

### Determination of Rab18 and TRAPPC9 expression in miR-1914-3p overexpressed macrophages

To overexpress miR-1914-3p mimic in THP-1 macrophages, $0.5 \times 10^6$ cells were plated in each well of a 6 well tissue culture plate and differentiated as mentioned earlier. After differentiation, cells were transfected with control or experimental miRNA mimic at the indicated concentrations using Lipofectamine RNAi MAX (Invitrogen) according to the manufacturer's instruction. Finally, level of Rab18 and TRAPPC9 were determined after 48 h by q PCR and western blot analysis as described above. Results were expressed as relative expression compared to the untreated control arbitrarily chosen as 1 unit.

### Determination of parasite survival in miR-1914-3p overexpressed macrophages

Differentiated THP-1 cells were transfected with miR-1914-3p (50 nM) or Rab18 specific siRNA (50 nM) using RNAi MAX transfection reagent (Invitrogen) as per manufacturer's protocol. Control siRNA transfected cells were used as control. Cells were infected with *Leishmania* promastigotes as described earlier [25,66] and chased for indicated periods of time. To understand the role of fatty acids in miR-1914-3p mediated inhibition of parasite growth in macrophages, infected macrophages transfected with miR-1914-3p were incubated in the medium containing fatty acids supplements like Linolenic acid-oleic acid-albumin (1x). At respective time point, coverslips containing infected cells were washed three times in PBS and fixed with methanol for 10 min at 24°C. Air dried coverslips were permeabilized with PBS containing 0.1% Triton X-100 and 0.1 mg/ml RNaseA for 1 h at 37°C. Samples were washed three times with PBS and stained for 30 min with 50 mg/ml of PI (Propidium iodide), followed by three washes with PBS. Coverslips were air dried and were mounted in ProLong gold antifade reagent (Molecular Probes) and viewed in Leica confocal microscope using an oil immersion 63X objective. Parasites load in the infected THP-1 macrophages were microscopically estimated and results are expressed as numbers of parasites present per 100 macrophages at the indicated time points.

### Statistical analysis

Statistical analysis was performed using Sigma Plot version 14.0. Student's two-tailed paired t test or two tailed Mann-Whitney test was used to determine differences between control and

test groups with >95% confidence intervals. Number and size of LBs were quantified by ImageJ software and graphs were plotted using Origin software. P values less than 0.05 was considered to be significant for all analyses.

## Supporting information

**S1 Fig. Demonstrates the recruitment of host Rab7, Rab8 or Rab27 on Ld-PVs in *L. donovani* infected and uninfected macrophages and respective quantitation.**
(TIF)

**S2 Fig. Demonstrates the transfection of miR-1914-3p or control mimic in macrophages.**
(TIF)

**S3 Fig. Demonstrates the recruitment of host Rab18 and LBs stained with LipidTOX on Ld-PVs in *L. donovani* infected and uninfected macrophages.**
(TIF)

**S4 Fig. Shows the inhibition of Rab18 expression by the overexpression of specific siRNA in transfected macrophages.**
(TIF)

**S5 Fig. Shows that the transfection of Rab18 specific siRNA or overexpression of miR-1914-3p specifically inhibits the recruitment of perilipin labelled LBs on Ld-PVs in macrophages.**
(TIF)

## Acknowledgments

We thank Dr. Syamal Roy of Indian Institute of Chemical Biology, Kolkata and Prof. Roop Mallik of Indian Institute of Technology, Bombay for critically reviewing the manuscript. We also acknowledge the support of Central Research Facility of Indian Institute Technology, New Delhi.

## Author Contributions

**Conceptualization:** Amitabha Mukhopadhyay.

**Data curation:** Jitender Kumar Verma, Rituparna Basak, Anjali Kapoor.

**Formal analysis:** Chandni Sood, Jitender Kumar Verma, Rituparna Basak, Anjali Kapoor.

**Funding acquisition:** Amitabha Mukhopadhyay.

**Investigation:** Chandni Sood, Jitender Kumar Verma, Rituparna Basak, Anjali Kapoor, Swarnima Gupta.

**Methodology:** Chandni Sood, Jitender Kumar Verma, Rituparna Basak, Anjali Kapoor, Swarnima Gupta.

**Project administration:** Amitabha Mukhopadhyay.

**Supervision:** Amitabha Mukhopadhyay.

**Writing – original draft:** Amitabha Mukhopadhyay.

**Writing – review & editing:** Amitabha Mukhopadhyay.

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
