## [Decision Letter · Decision Letter 0]

15 Aug 2023

Dear professor Mukhopadhyay,

Thank you very much for submitting your manuscript "Leishmania highjack host lipid body to acquire fatty acids by overexpressing Rab18 and TRAPPC9 by downregulating miR-1914-3p expression in macrophages" for consideration at PLOS Pathogens. As with all papers reviewed by the journal, your manuscript was reviewed by members of the editorial board and by several independent reviewers. In light of the reviews (below this email), we would like to invite the resubmission of a significantly-revised version that takes into account the reviewers' comments.

All three reviewers found the manuscript to be potentially novel, but felt that there were major issues with the manner in which experimental data was presented and interpreted. I draw attention to a need for more rigour in data presentation, including ensuring that all necessary controls are present, in interpreting data in Results sections (taking care to avoid any over-interpretation), and in carefully explaining caveats in the Discussion. One reviewer also felt that PV identification was often not unequivocal. These issues should be thoroughly addressed in any revised manuscript.

We cannot make any decision about publication until we have seen the revised manuscript and your response to the reviewers' comments. Your revised manuscript is also likely to be sent to reviewers for further evaluation.

Sincerely,

Ashraful Haque

Academic Editor

PLOS Pathogens

Kami Kim

Section Editor

PLOS Pathogens

Kasturi Haldar

Editor-in-Chief

PLOS Pathogens

orcid.org/0000-0001-5065-158X

Michael Malim

Editor-in-Chief

PLOS Pathogens

orcid.org/0000-0002-7699-2064

All three reviewers found the manuscript to be potentially novel, but felt that there were major issues with the manner in which experimental data was presented and interpreted. I draw attention to a need for more rigour in data presentation, including ensuring that all necessary controls are present, in interpreting data in Results sections (taking care to avoid any over-interpretation), and in carefully explaining caveats in the Discussion. One reviewer also felt that PV identification was often not unequivocal. These issues should be thoroughly addressed in any revised manuscript.

Reviewer's Responses to Questions

**Part I - Summary**

Reviewer #1: The manuscript by Sood et al. “Leishmania highjack host lipid body to acquire fatty acids by overexpressing Rab18 and TRAPPC9 by downregulating miR-1914-3p expression in macrophages” sets out to dissect the mechanism of fatty acid uptake from the host by the kinetoplastid parasite Leishmania donovani.

Since Leishmania is a major human pathogen and serves as a valuable cell biological model understanding the uptake mechanism of fatty acids is significant from an intervention strategy and from a general knowledge perspective.

Through a set of experiments (microscopy, western blot, semiquantitative PCR) the authors conclude that dicer is degraded by gp63, which reduces the expression of miR-1914-3p. This in turn upregulates Rab18 and TRAPPC9, which induces recruitment of fatty acid containing lipid bodies to the parasitophorous vacuole. This set of events would provide a significant, comprehensive insight into the mechanism of fatty acid uptake by Leishmania donovani.

However, the manuscript lacks scientific rigour and suffers from over-interpretation (see major issues).

Reviewer #2: Sood and colleagues investigated how Leishmania modulates the host cell traffic to get the lipid bodies in the parasitophorous vacuole. First, they observed the increased expression of Rab18 and its GEF, TRAPPC9, in L. donovani-infected macrophages. Rab18 is a well-known small GTPase involved in intracellular traffic, especially of lipid bodies. As miRNAs are imputed as major post-transcriptional regulators, they compared their expression in infected and uninfected cells. Among the reduced miRNAs was mi-1914-3p, which was found to be complementary to Rab18 and GEF RNAs domains. This regulation loop was well demonstrated in the MS. Following, they showed the increase of biogenesis and recruitment of lipid bodies by the parasitophorous vacuole. The authors also showed that the parasite metalloprotease GP63, when transfected in HeLa cells, reduced the expression of Dicer. The final finding of the MS is the demonstration that modulating the levels of mi-1914-3p or Rab18 directly interferes with fatty acids colocalization with parasitophorous vacuoles. They correlate the lower availability of fatty acids with the minor development of amastigotes (what they call parasite death), although they do not establish a relationship between these two events. The MS is well-designed, has a good rationale, and addresses an exciting cell biology question.

Reviewer #3: In this fascinating new manuscript, the focus on the connection between lipid bodies (or lipid droplets) ad the PV of Leishmania Donovani. The authors use gene expression analysis, confocal microscope and manipulation of expression of Rab18 and miR-1914-3p to conclude that Leishmania manipulated Dicer2 levels, which in turn control miR-1914-3p levels, increase Rab18 ant TRACC9 to deliver LBs to the PV.

**Part II – Major Issues: Key Experiments Required for Acceptance**

Reviewer #1: The manuscript relies mainly on two lines of evidence: the up- or down-regulation of molecules and the localisation of molecules depending on infection status. Most experiments are not explained in sufficient details and lack appropriate controls:

- Some of the bands in the western blots (especially loading controls) are already in saturation introducing artefacts and making quantification unreliable. This increase in expression is also not necessarily supported by the provided microscopy pictures.

-PFA potentially destroys surface structures and hence the conclusion that live Leishmania are necessary cannot be drawn from this experiment.

- Microscopy data:

>No correlation between the structures detected by Rab18 and the LipidTOX stain is shown. No data on the acquisition is provided (how many cells?, settings?, were the same setting used?...).

>Scale bars are only just visible when one zooms in. PV is never shown (DIC image or specific marker), but drawn onto the figure – this is an interpretation by the authors and not primary data that can be evaluated.

>Given the very weak signal of Trappc9 in uninfected cells, it would be good to show specificity and controls using only secondary antibodies.

>Fig. 4A - no nuclei visible in uninfected host cell, raises question of consistency of data/setting/robustness, again quantification murky, not enough detail provided to interpret the data, recruitment or non-recruitment of LB is not obvious

>Fig. 4D –5B “lipid markers nearby PV” – what does “nearby mean”? What were the objective criteria for this analysis?

>Fig. 4E - How where the number of lipid bodies determined (Only one section or were z-stacks produced?) According to the text, 50 macrophages were analysed (per condition?), but the figure only shows 5 data points.

>Fig. 4G - “In addition, we found that calnexin-labelled ER membrane is also recruited on PVs indicating plausible mechanism how newly synthesized LBs bearing ER membrane are recruited on PVs (Fig.4g).” This is a throw-away comment – no analysis of this data is provided nor is this explored in any detail.

- In general, Figure legends and Material and Methods lack important information and do not allow for the evaluation of the data nor for its reproducibility (e.g. Fig. 3f – incubation time for gp63).

- It is hard to follow the logic of the study: often it is not obvious, why a particular experiment was undertaken as a next step. This becomes clearer in the discussion, where the authors outline their chain of argument, however, they often use circumstantial rather than actual evidence (e.g. by citing situations from unrelated organisms).

- The manuscript would benefit from a more cautious interpretation of the data (expecially given the experimental shortcomings listed above). The discussion is spiked with statements as "To unequivocally prove..", "we have clearly demonstrated...", "we have precisely addressed how..." or "Toxoplasma is the only other intracellular parasite..." (Have ALL other intracellular parasites been investigated?).

- Often there is a functional link implied, where there is no evidence provided: e.g. "The enhance recruitment of Rab18 and its GEF on PV facilitates the docking of LB on PV and thereby parasites acquire fatty acids from LBs for their survival." (no acquisition of fatty acids was shown)

or

"Our results also indicate that overexpressions of Rab18 and TRAPPC9 induce the biogenesis of LBs in infected cells as biogenesis of LBs in cells are shown to be predominantly regulated by Rab18 and its GEF [10, 11].” (Rab18 and GEF have been identified in these publications as important (but not prddominant) factors and biogenesis of LBs has not been shown in the submitted study.)

Reviewer #2: Although I do not think it is necessary to conduct additional experiments, the presented data do not support some conclusions, so they must be reformulated, and I considered them major issues.

1 – The title (“Leishmania highjack host lipid body to acquire fatty acids…”) and the justificative in the introduction point the fatty acids acquisition as the main question to be addressed in the manuscript, but, actually, what is shown is the colocalization of lipid bodies and parasitophorous vacuoles. No acquisition of fatty acids by parasites is evaluated. Although fatty acids are significant constituents of lipid bodies, other molecules, such as cholesterol, phospholipids, and even some proteins, are present. I suggest reformulating the title.

2 - In the first sentence of the author summary, it is stated that “Leishmania donovani is unable to synthesize fatty acids and depends on its host to scavenge it for its survival”. The fatty acid synthesis in Leishmania has already been described. For a recent review, see Arya R, Dhembla C, Makde RD, Sundd M, Kundu S. An overview of the fatty acid biosynthesis in the protozoan parasite Leishmania and its relevance as a drug target against leishmaniasis. Mol Biochem Parasitol. 2021 Nov;246:111416. doi: 10.1016/j.molbiopara.2021.111416. Epub 2021 Sep 21. PMID: 34555376.

Please, revise this throughout the entire text.

3 – Please, see these detached sentences from the manuscript:

- Abstract: “Moreover, overexpression of miR-1914-3p in macrophages significantly inhibits the recruitment of LBs and thereby suppresses the growth of parasites in macrophages as parasites are unable to acquire fatty acids.”

- Author summary: “We have also shown that overexpression of miR-1914-3p in macrophages significantly inhibits the recruitment of LBs and thereby suppresses the growth of parasites in macrophages as parasites are unable to acquire fatty acids.”

- Results: “To determine whether Rab18 function is necessary for acquiring fatty acids from LBs in the survival of Leishmania in macrophages…”

“These results indicated that parasite acquires fatty acid from LBs for its survival.”

- Discussion: “These results demonstrate that overexpression of Rab18 by downregulating the expression of miR-1914-3p by Leishmania is essential to acquire FAs from host LBs for the survival of parasites in macrophages.”

The experimental approach was to modulate the levels of mi-1914-3p or Rab18 and to evaluate the localization of fatty acids derived from lipid bodies and the number of amastigotes in each case. So, several points are raised from the above sentences:

- The parasite viability was not accessed, just its proliferation over time. Then, it sounds too strong mentioning parasite survival.

- The fatty acid incorporation by the parasite was not addressed at any time in the manuscript, just its availability from the lipid bodies (other possible pathways, such as LDL uptake, were not evaluated) in the parasitophorous vacuole. So, again sounds too strong the statement “thereby suppresses the growth of parasites in macrophages as parasites are unable to acquire fatty acids”.

- miR1914-3p and Rab18 are involved in other cell processes. Then, it is not possible to state that parasite growth is impaired due to the lack of fatty acids from the lipid bodies. No reversion experiments were conducted, such as fatty acid supplementation.

Please, revise the text in light of the discussion above.

4 – Page 6 – “Consequently, we showed that gp63, a metalloprotease in Leishmania, degraded the Dicer in HeLa cells”. The data presented are related to the reduced dicer expression in the transfected cells. It was not shown whether gp63 is able to degrade dicer directly, nor how this enzyme would be in contact with dicer inside the cell.

5 – Page 6 – “These results demonstrated that Leishmania degrades the Dicer by its gp63 and thereby reduces the expression of miR-1914-3p…”. The only presented experiment involving gp63 was transfecting HeLa cells with the metalloprotease and evaluating the dicer expression. No relationship among gp63, dicer degradation, and expression of miR-1914-3p was stablished.

Reviewer #3: 3 Primary issues stand out:

1. Most the imaging does NOT unequivocally identify the PV, with the exception of Figure 5 that includes LAMP1 staining that outlines the PVs. In all the other figures, it is not possible to accurately identify the PV and determine if the conclusions about association of PV with LB and/or Rab18 are accurate. This needs to be thorough addressed with. new data. It would also be an significant improvement if the Leishmania parasite were visualized by something more specific then small little dotes of PI or Draq5 stain.

2. The authors claim to "unequivocally demonstrate" their model, however (even ignoring the issue above) the data as a whole are consistent with the conclusion but do not prove/show/demonstrate the model. The authors need to find a way to interfere with their proposed pathway: GP63-|Dicer-|miR-1914-3p-|Rab18 and LB recruitment to the PV. Artificially over-expressing miR-1914-3p nor knocking down Rab18 are not the same as blocking this pathway; these experiments do no provide unequivocal demonstration. Moreover, Figure 5 that aims to demonstrate fatty acid recruitment to the PV is not convincing. Seems like there is more FA signal in the general vicinity of the PVs, as seen in the image of the whole cells, but it is not clear from the inset/zoom in that actual fusion or delivery of the FA to the PV membrane, and certainly not into the PV, even at 24hr has occurred. Also, the time course shown in Figure 5A is critically important for the conclusion of this paper, but only 1 cells is shown for each time point and there is no quantitation.

3. Another important issue, but more about language, is that the authors often write that Rab18 or TRACC is "overexpressed" in the context of Leishmania infection. This has the connotation of experimental over expression, for example with transfection of an expression plasmid. However, in fact, this is a mild (about 2x fold according to the data presented) increase in protein and mRNA expression observed with Leishmania infection. Different language should be used to make this point more clearly. Moreover, it raises a question: is this 2x fold change in Rab18 (and TRACC9) really sufficient drive this dramatic change in LB localization?

**Part III – Minor Issues: Editorial and Data Presentation Modifications**

Reviewer #1: - Often it is not clear whether the authors talk about host or parasite molecules (gp63, Rab18, TRAPPC9,…).

- If the degradation of miRNA is indeed via dicer, how come that only 40 miRNA are affected – one would expect a more global impact? (The authors themselves ask the questions why there are also miRNAs that are upregulated.)

-Discussion:

A figure outlining the proposed mechanism would be useful.

-The text states “Leishmania induced more than 2.5-folds expression of Rab18”, whereas neither quantification in Fig. 1a or 1b shows this.

- The manuscript needs to be checked for grammar - in particular the use of articles, plural forms and the use of "like/etc.".

- The title, abstract, introduction and discussion imply that this study is relevant to all Leishmania species, however, technicallly only Leishmania donovani was investigated.

- Introduction of abbreviations e.g. GEF

Reviewer #2: Page 10: “Leishmania do not have complete lipid synthesis pathway and therefore, parasite needs to scavenge lipids from the host cells [43].” This sentence is vague (what would be a complete lipid synthesis?), and the reference is misused. This reference does not address this question. This sentence repeats below in text.

Reviewer #3: Almost all the figures are spread across multiple pages. This is not usually acceptable at most journals.

PLOS authors have the option to publish the peer review history of their article (what does this mean?). If published, this will include your full peer review and any attached files.

Reviewer #1: No

Reviewer #2: No

Reviewer #3: No
---

## [Decision Letter · Decision Letter 1]

22 Jan 2024

Dear professor Mukhopadhyay,

Thank you very much for submitting your manuscript "Leishmania highjack host lipid body to acquire fatty acids by overexpressing Rab18 and TRAPPC9 by downregulating miR-1914-3p expression in macrophages" for consideration at PLOS Pathogens. As with all papers reviewed by the journal, your manuscript was reviewed by members of the editorial board and by several independent reviewers. The reviewers appreciated the attention to an important topic. Based on the reviews, we are likely to accept this manuscript for publication, providing that you modify the manuscript according to the review recommendations.

Please adjust interpretation of your Results in reference to concerns raised by Reviewer 2, in particular by inserting caveats and possible alternative explanations for your data. In situations were causality can not be demonstrated, it will be important to acknowledge this. Please also address the minor issues from the other Reviewer.

Sincerely,

Ashraful Haque

Academic Editor

PLOS Pathogens

Kami Kim

Section Editor

PLOS Pathogens

Kasturi Haldar

Editor-in-Chief

PLOS Pathogens

orcid.org/0000-0001-5065-158X

Michael Malim

Editor-in-Chief

PLOS Pathogens

orcid.org/0000-0002-7699-2064

Please adjust interpretation of your Results in reference to concerns raised by Reviewer 2, in particular by inserting caveats and possible alternative explanations for your data. In situations were causality can not be demonstrated, it will be important to acknowledge this. Please also address the minor issues from the other Reviewer.

Reviewer Comments (if any, and for reference):

Reviewer's Responses to Questions

**Part I - Summary**

Reviewer #2: (No Response)

Reviewer #3: Same as prior review

**Part II – Major Issues: Key Experiments Required for Acceptance**

Reviewer #2: Question 2: The title (“Leishmania highjack host lipid body to acquire fatty acids…”) and the justificative in the introduction point the fatty acids acquisition as the main question to beaddressed in the manuscript, but, actually, what is shown is the colocalization of lipid bodies and parasitophorous vacuoles. No acquisition of fatty acids by parasites is evaluated. Although fatty acids are significant constituents of lipid bodies, other molecules, such as cholesterol, phospholipids, and even some proteins, are present. I suggest reformulating the title.

Reply 2: I would like to submit that we have used BODIPY™ 493/503 and LipidTOX™ to stain the LBs and anti-perilipin antibody is used to detect lipid droplet-associated protein. Thus, these reagents stained the Lipid bodies and used for the recruitment of LBs by LdPV as agreed by the reviewer. However, BODIPY-FL-C12 is a saturated FA analog that specifically labels the fatty acid present within the lipid bodies. Therefore, transfer of BODIPY-FL-C12 from Lipid bodies to the target organelles is used to determine the trafficking of fatty acids in various studies by pulse-chase experiments [Developmental Cell (2015) 32, 678–692; Cell Metabolism (2018) 27, 886–897]. We have used the same well-established method to determine fatty acid acquisition by LdPV using FLC12-labelled FAs in LBs by pulse-chase experiments. I think title is appropriate as we have shown the fatty acid recruitment by LdPV. Moreover, we have added a new figure in the revised manuscript showing that the addition of fatty acids supplement reversed the inhibition of parasite growth in mi-1914-3p treated macrophages indicating that parasites acquire fatty acids by enhanced expression of Rab 18 via the downregulation of mi-1914-3p in infected macrophages. Therefore, we believe that title is appropriate. Moreover, this title is also agreed by other reviewers.

Reviewer reply: The authors assume that the mere access of fatty acids to the PV automatically results in their incorporation into the parasites, which cannot be true. They did not demonstrate through any technique that the labeled fatty acid is inside the amastigotes. Furthermore, the fact that BODIPY-FL-C12 has been found inside the PV does not rule out the involvement of other constituents of LBs. In other words, the presented data show that fatty acids from LBs are in the PV, but this does not necessarily mean that they are incorporated by the parasites, nor does it imply that they are the only significant molecules from LBs relevant to the observed phenomenon in the sequence.

Additionally, the fact that the other reviewers have not mentioned the title in their reviews does not necessarily mean that they agree.

Question 5: The experimental approach was to modulate the levels of mi-1914-3p or Rab18 and to evaluate the localization of fatty acids derived from lipid bodies and the number of amastigotes in each case. So, several points are raised from the above sentences: - The parasite viability was not accessed, just its proliferation over time. Then, it sounds too strong mentioning parasite survival.

Reply 5: I would like to submit that if parasite does not proliferate in macrophages which means that parasites in not surviving in host cells. However, we have used proliferation/multiplication of the parasites in the revised manuscript as indicated by the reviewer.

Reviewer comment: Thank you for adjusting the text. Just for knowledge, it has been well stablished the quiescence phenomenon in Leishmania. Please, see the following papers:

Jara M, Barrett M, Maes I, Regnault C, Imamura H, Domagalska MA, Dujardin JC. Transcriptional Shift and Metabolic Adaptations during Leishmania Quiescence Using Stationary Phase and Drug Pressure as Models. Microorganisms. 2022 Jan 3;10(1):97. doi: 10.3390/microorganisms10010097.

Kloehn, J.; Saunders, E.; O’Callaghan, S.; Dagley, M.J.; McConville, M.J. Characterization of Metabolically Quiescent Leishmania Parasites in Murine Lesions Using Heavy Water Labeling. PLoS Pathog. 2015, 11, e1004683

Question 6: The fatty acid incorporation by the parasite was not addressed at any time in the manuscript, just its availability from the lipid bodies (other possible pathways, such as LDL uptake, were not evaluated) in the parasitophorous vacuole. So, again sounds too strong the statement “thereby suppresses the growth of parasites in macrophages as parasites are unable to acquire fatty acids”.

Reply 6: I am sorry to state that the reviewer has misinterpreted our data with BODIPY-FLC12 which specifically labelled the fatty acids. We have used a well-established method to determine transfer of fatty acids using FLC12-labelled FAs [Developmental Cell (2015) 32, 678–692; Cell Metabolism (2018) 27, 886–897].

Reviewer reply: I regret to note that the authors may not fully grasp that the colocalization of a molecule in the PV does not constitute proof that the molecule is inside the parasite. Furthermore, this does not rule out the possibility of the involvement of other LB constituents, nor does it exclude other potential sources of fatty acids, as mentioned.

Question 8: Page 6 – “Consequently, we showed that gp63, a metalloprotease in Leishmania, degraded the Dicer in HeLa cells”. The data presented are related to the reduced dicer expression in the transfected cells. It was not shown whether gp63 is able to degrade dicer directly, nor how this enzyme would be in contact with dicer inside the cell. Page 6 – “These results demonstrated that Leishmania degrades the Dicer by its gp63 and thereby reduces the expression of miR-1914-3p…”. The only presented experiment involving gp63 was transfecting HeLa cells with the metalloprotease and evaluating the dicer expression. No relationship among gp63, dicer degradation, and expression of miR-1914-3p was stablished.

Reply 8: We have revised the text in light of the reviewer suggestions. Reviewer has stated that “No relationship among gp63, dicer degradation, and expression of miR-1914-3p was established”. However, we have shown in the Fig. 3f of the reviewed manuscript that the expression of gp63 alone reduces the expression of dicer. Moreover, I am sorry to inform that this is in sharp contrast with Reviewer 1 statement where he has said that “Through a set of experiments authors have shown that dicer is degraded by gp63, which reduces the expression of miR-1914-3p. This in turn upregulates Rab18 and TRAPPC9, which induces the recruitment of fatty acid-containing lipid bodies to the parasitophorous vacuole. Accordingly, reviewer 1 has commented that “This set of events provide a significant, comprehensive insight into the mechanism of fatty acid uptake by Leishmania donovani”. The same is also agreed by the Reviewer 3.

Reviewer reply: I would like the authors to specify where they demonstrate the degradation of Dicer by GP63. I was unable to find the Western blot illustrating the degradation of Dicer, as well as an explanation of how a major surface glycoprotein can gain access to Dicer within the parasite.

Reviewer #3: The first round of critiques have been thoroughly addressed. Major issues are addressed with new data, especially LAMP1 staining throughout and also the FA supplementation experiment.

In addition, the language has been adjusted so that conclusions and interpretations are not overstated.

One improvement that should be easy, but would be a major improvement: in figure 5A, can the "acquired" FA be noted with an arrow or similar...what are the authors scoring as "acquired". This was a key issue in the prior reviews and could be clarified further, although the data is all there, I believe.

**Part III – Minor Issues: Editorial and Data Presentation Modifications**

Reviewer #2: (No Response)

Reviewer #3: Be sure all graphs are completely labeled. For example, what are the different sets of bars in 4d? The x-axis is not labeled.

I think there is a stray Asterix and P value noted on the lower half of 4e (the main text says no difference).

5c graph should have x-axi labels too...presumably different time points.

PLOS authors have the option to publish the peer review history of their article (what does this mean?). If published, this will include your full peer review and any attached files.

Reviewer #2: No

Reviewer #3: No

Figure Files:

Data Requirements:

Reproducibility:

References:

---

## [Editor Report · Decision Letter 2]

5 Feb 2024

Dear professor Mukhopadhyay,

We are pleased to inform you that your manuscript 'Leishmania highjack host lipid body for its proliferation in macrophages by overexpressing host Rab18 and TRAPPC9 by downregulating miR-1914-3p expression' has been provisionally accepted for publication in PLOS Pathogens.

Best regards,

Ashraful Haque

Academic Editor

PLOS Pathogens

Kami Kim

Section Editor

PLOS Pathogens

Michael Malim

Editor-in-Chief

PLOS Pathogens

orcid.org/0000-0002-7699-2064
---

## [Editor Report · Acceptance letter]

12 Feb 2024

Dear professor Mukhopadhyay,

We are delighted to inform you that your manuscript, "Leishmania highjack host lipid body for its proliferation in macrophages by overexpressing host Rab18 and TRAPPC9 by downregulating miR-1914-3p expression," has been formally accepted for publication in PLOS Pathogens.

Best regards,

Michael Malim

Editor-in-Chief

PLOS Pathogens

orcid.org/0000-0002-7699-2064